# Electrospun Nanofiber Membrane for Cultured Corneal Endothelial Cell Transplantation

**DOI:** 10.3390/bioengineering11010054

**Published:** 2024-01-05

**Authors:** Euisun Song, Karen M. Chen, Mathew S. Margolis, Thitima Wungcharoen, Won-Gun Koh, David Myung

**Affiliations:** 1Department of Ophthalmology, Spencer Center for Vision Research, Byers Eye Institute at Stanford University, Palo Alto, CA 94304, USA; 2VA Palo Alto Health Care System, Palo Alto, CA 94304, USA; 3Department of Chemical and Biomolecular Engineering, Yonsei University, 50 Yonsei-Ro, Seodaemun-Gu, Seoul 120-749, Republic of Korea; 4Ophthalmology, MedPark Hospital, Bangkok 10110, Thailand; 5Department of Chemical Engineering, Stanford University, Stanford, CA 94305, USA

**Keywords:** electrospinning, gelatin, nanofiber, corneal endothelial cell, cornea transplantation

## Abstract

The corneal endothelium, comprising densely packed corneal endothelial cells (CECs) adhering to Descemet’s membrane (DM), plays a critical role in maintaining corneal transparency by regulating water and ion movement. CECs have limited regenerative capacity within the body, and globally, there is a shortage of donor corneas to replace damaged corneal endothelia. The development of a carrier for cultured CECs may address this worldwide clinical need. In this study we successfully manufactured a gelatin nanofiber membrane (gelNF membrane) using electrospinning, followed by crosslinking with glutaraldehyde (GA). The fabricated gelNF membrane exhibited approximately 80% transparency compared with glass and maintained a thickness of 20 µm. The gelNF membrane demonstrated desirable permeability and degradability for a Descemet’s membrane analog. Importantly, CECs cultured on the gelNF membrane at high densities showed no cytotoxic effects, and the expression of key CEC functional biomarkers was verified. To assess the potential of this gelNF membrane as a carrier for cultured CEC transplantation, we used it to conduct Descemet’s membrane endothelial keratoplasty (DMEK) on rabbit eyes. The outcomes suggest this gelNF membrane holds promise as a suitable carrier for cultured CEC transplantation, offering advantages in terms of transparency, permeability, and sufficient mechanical properties required for successful transplantation.

## 1. Introduction

The cornea must remain transparent to ensure clear vision. It comprises five layers: three cell-laden layers and two made primarily of proteins. The corneal endothelium, the innermost layer of the cornea, comprises a single layer of corneal endothelial cells (CECs) maintaining a high cell density. It is anchored to Descemet’s membrane (DM) which is approximately 10 µm in thickness. Human CECs are inherently unable to proliferate within the body and are thus incapable of regeneration in the event of cell loss [1]. The paramount role of CECs is to transport nutrients from the anterior chamber while maintaining precise hydration levels, thereby preserving the thickness and transparency of the cornea [2,3]. Disruptions in this function results in edema of the corneal stoma and surface epithelium [4,5]. Given the global scarcity of corneal donations, there remains an urgent need for the development of cultured CEC transplantation methods. Current methods for transplanting CECs include direct injection and transplantation as sheets on fabricated carriers, mimicking DMEK transplant surgery. Direct cell injection has the advantage of minimizing immune responses since it does not require additional materials [6]. While direct cell injection approaches have shown promising results in clinical trials [7,8], they are currently not yet FDA-cleared and widely available for use in patients. A need remains to develop transplantation methods that overcome the severe shortage of graft tissue worldwide. This may be achieved by creating a minimally immunogenic transient carrier to deliver cells to the inner cornea. The material used for this purpose should be thin and flexible to facilitate scrolling and folding during surgery, porous to facilitate permeability, possess appropriate mechanical properties for surgical handling, and be transparent.

Electrospinning is one of the most valuable tools for creating thin membranes with high permeability. It is capable of producing various forms of nanofiber scaffolds and has been actively utilized in the field of tissue engineering, particularly in wound healing, vascular grafts, and cardiac tissue engineering [9]. In a DMEK corneal transplantation procedure, a 7 to 8 mm circular disc of Descemet’s membrane and attached endothelial cells are rolled into a cylinder before they are injected into the anterior chamber through a 2.4–4.0 mm incision. The transplant graft is then unfurled and properly placed endothelial side down before adhering to the posterior cornea with the aid of a gas bubble. DMEK is currently employed successfully by corneal surgeons with minimal endothelial cell loss. To replicate this process with cultured CECs, an engineered membrane must properly mimic Descemet’s membrane. Carriers primarily developed using highly biocompatible natural polymers tend to lose their mechanical properties during cell-culturing processes, rendering them unsuitable for transplantation.

In this study, we aimed to develop a carrier for CEC transplantation by electrospinning a gelatin solution into nanofiber membranes. Gelatin, derived from collagen, has been applied across various disciplines, including wound healing and bone regeneration [10,11], leveraging its notable biocompatibility. Furthermore, it manifests characteristics conducive to electrospinning, such as heightened solubility in non-organic solvents. This property not only mitigates toxicity but also allows for facile modification of the solvent in the preparation of electrospinning solutions. Our objective was to minimize toxicity and immune reactions by reducing the use of other synthetic polymers and organic solvents. gelNF membranes have been utilized in various fields, but to achieve enhanced properties, they are often blended with a variety of synthetic polymers [12,13]. To address challenges related to transparency and toxicity issues associated with organic solvent use, we sought to explore the potential use of electrospun gelNF membranes as carriers for CEC transplantation. After confirming protein expression via the transplantation of human and rabbit CECs onto the fabricated gelNF membrane, we conducted ex vivo tests to validate the transplantation of cells into freshly harvested rabbit corneas.

## 2. Materials and Methods

### 2.1. Fabrication of Electrospun Gelatin Nanofiber (gelNF)

To prepare the gelatin solution for the electrospinning, gelatin type A (G1890 Sigma Aldrich, St. Louis, MO, USA) was dissolved in a 70% acetic acid solution (695092, Sigma Aldrich) and 30% distilled water. The gelatin solution (25%, *w*/*v*) was then placed into a 5 mL syringe and connected to a 23-gauge needle. The electrospinning machine (TL-01, Tong Li Tech) fabricated the nanofiber membrane using a flow rate of 1 mL/h, a 17 kV voltage, and a 15 cm distance from the needle to the collector. The nanofibers were collected on the cover glass, covered with aluminum foil, and evaporated overnight at RT to remove residual solvent. The samples were then crosslinked with glutaraldehyde (G6257, Sigma Aldrich) in a vapor crosslinking system and evaporated overnight at RT to remove residual solvent. The nanofiber membranes were soaked in PBS and then separated from the foil. The fabricated samples were sterilized under UV for 30 min for in vitro and ex vivo studies.

### 2.2. Characterization of gelNF Membrane

To characterize the properties of the gelNF membrane, thickness light transmittance, permeability, degradability, and tensile strength were assessed. To measure the light transmittance, the gelNF membranes were cut into a 96-well size using punch biopsy and placed into a plate reader (Spark^®^, Tecan Life Sciences, Männedorf, Switzerland) at the wavelength of visible light. The transparency of each gelNF membrane was evaluated in comparison with a cover slip with a reference transparency of 100%. To analyze the morphology of the nanofibers and calculate the diameter of the nanofibers, scanning electron microscopy (SEM, Thermo Fisher Scientific Apreo, Hillsboro, OR, USA, n = 100) was conducted. The fiber diameter was calculated from 100 fibers in each group using image J (Version 1.54h). The thickness of the gelNF membranes was measured using a vernier caliper (Fowler, Valencia, CA, USA, n = 3) and SEM. FITC–dextran (FD10S and FD70, Sigma Aldrich) and cell crowns (CellCrown^TM^, Scaffdex, Tampere, Finland) were used to test the permeability (n = 3). First, 12.5 μg/mL of the dextran dissolved in the cell culture medium was added at the top of the cell crown. After 24 h of incubation, the medium from the bottom part of the cell crown was measured to calculate the penetrated dextran. For degradability testing, the gelNF membranes were incubated at 37 °C in PBS for 14 and 28 days, and SEM was employed to evaluate changes in the nanofibers. To measure the tensile strength of the membranes, an Instron testing instrument (INSTRON 5560, Norwood, MA, USA, n = 9) was used, and the specimens were prepared with a rectangular shape (5 × 5 cm). The 30 mm/min strain was assessed with a 1 kN load cell until breakpoint. The elastic modulus was calculated from the stress–strain curve.

### 2.3. Immortalized Human Corneal Endothelial Cell (IhCEC) Culture

The IhCEC (T0577) was purchased from Applied Biological Materials (ABM, Richmond, BC, Canada) and cultivated following the manufacturer's protocol. Briefly, the culture medium (TM001) was supplemented with 10% FBS (Gibco), 5 μg/mL of human insulin (TM058), 10 µg/mL of human transferrin, 3 ng/mL of sodium selenite, 10 nM of hydrocortisone, 10 nM of β-estradiol, 10 ng/mL of rhVEGF 165aa (Z100895), 10 ng/mL of rhEGF (Z100135), 10 ng/mL of heparin, 2 mM of L-glutamine (G275), and a 1% penicillin/streptomycin solution (15140122, Gibco). The IhCECs were cultivated on a substrate coated with FNC coating mix (Athena (0407, Athena)) at 37 °C in a humidified incubator with 5% CO_2_ and subcultured using 0.05% trypsin–EDTA (Gibco). The culture medium was changed every 2–3 days.

### 2.4. Isolated Primary Rabbit Corneal Endothelial Cell (PrCEC) Culture

Rabbit eyeballs were purchased from Visiontech Inc. (Sunnyvale, TX, USA). To isolate the CECs, Descemet’s membrane was separated and placed in a 2 mg/mL collagenase (C0130, Sigma Aldrich) solution to incubate at 37 °C for 1 h. It was centrifuged at 1000× *g* rpm for 3 min and resuspended with FNC coating mix, Opti-MEM, and supplements (11058021, Gibco). To analyze the elongation ratio of the PrCECs, we drew horizontal and vertical lines meeting perpendicularly within the cell membrane and compared the length ratios of these two lines. To culture the CECs on top of the gelNF membrane, the cells were seeded at a density of 8 × 10^4^ cm^2^. A phase-contrast microscope (EVOS, Thermo Fisher Scientific) was used for the analysis of the morphology of the CECs.

### 2.5. Cytotoxicity of the gelNF Membrane and Proliferation of CECs Cultured on Top of the gelNF Membrane

To evaluate the cytotoxicity of the gelNF membrane, Cell Counting Kit-8 (CK04, Dojindo) was performed following the manufacturer’s protocol (n = 6). Briefly, 10% of the working reagents' total volume in the culture medium was added and incubated for 90 min at 37 °C. An amount of 100 μL was transferred into a 96-well plate for absorbance measurements at a 450 nm wavelength. To test the viability of the CECs cultured on top of the gelNF membrane, we performed a Live & Dead assay kit following the manufacturer’s protocol. Briefly, the CECs were cultured on top of the gelNF membrane for 24 h. The culture medium was then changed to a Live & Dead solution mixed with a fresh culture medium. After a 15 min incubation, the live and dead cells were observed via a confocal microscope. The live cell ratio was determined by dividing the number of live cells by the total cell count (n = 5).

### 2.6. Ex Vivo Study of Cultured CEC Transplantation Using Artificial Anterior Chamber

The rabbit corneas (n = 5) were dissected from the eyeballs, followed by Descemet’s membrane stripping to remove the endothelia. An artificial anterior chamber (K20-2125, Corza Medical, Westwood, MA, USA) was utilized to mimic the eyeball shape. Three points were marked to distinguish the direction of cell cultivation: one point in the 12 o’clock direction and two points in the 2 o’clock direction on the gelNF membrane. The rabbit CECs were seeded at a density of 8 × 10^4^/cm^2^ and cultured on the gelNF membrane until they became confluent and were prepared into a circular shape with a diameter of 8 mm using a biopsy punch. To transplant the gelNF membrane, an incision was made using a 2.5 mm (8065921501, Alcon) knife, followed by injection into the anterior chamber using a glass Geuder (CorneaGen). Subsequently, an air bubble was introduced using a cannula and allowed to wait for 5 min. After removing the air bubble, the cell culture medium was added. After 3 days of incubation, the cells and gelNF membrane were examined using a confocal microscope.

### 2.7. Immunofluorescence Staining

To investigate the protein expressions in the CECs, the samples were rinsed with phosphate-buffered saline (PBS, Thermo Fisher Scientific, Waltham, MA, USA) and fixed using 4% paraformaldehyde (15710, Electron Microscopy Sciences, Hatfield, PA, USA) for 30 min at RT followed by washing 3 times using PBS. The samples were then permeabilized using 0.2% triton X-100 (93443, Sigma Aldrich) for 5 min followed by washing 3 times with PBS. The samples were subsequently incubated with 3% bovine serum albumin (BSA, A2153, Sigma Aldrich) dissolved in PBS for 1 h at RT and then incubated in a primary antibody dissolved in a 1% BSA solution overnight at 4 °C. The following day, the samples were rinsed 3 times with PBS and incubated with a secondary antibody dissolved in a 1% BSA solution for 1 h at RT. For DAPI staining, the DAPI solution was diluted in PBS and incubated for 5 min at RT. The images were photographed using a confocal microscope (LSM T-PMT, ZEISS, Jena, Germany). The primary and secondary antibodies used in this study were ZO-1(339188, Thermo Fisher Scientific), Na^+^/K^+^-ATPase (sc-48345, Santa Cruz Biotechnology, Dallas, Texas, USA), N-cadherin (ab98952, Abcam), aquaporin-1 (ab219055, Abcam, Cambridge, UK), phalloidin-555 (ab176756, Abcam), and DAPI (62248, Thermo Fisher Scientific).

### 2.8. Statistical Analysis

All the data are presented as means ± standard deviation. Statistical analysis was performed using a two-tailed (α = 0.05) Student's t-test for two experimental groups. One-way ANOVA with a post hoc Tukey’s multiple comparison test was applied for more than three test groups. A statistically significant difference was denoted as ** (*p* < 0.01) or *** (*p* < 0.001).

## 3. Results

### 3.1. Characterization of gelNF Membrane

First, we established the electrospinning conditions to impart the necessary characteristics for utilizing the produced gelNF membrane as a corneal material. The transparency of the gelNF membrane, measured in the visible light wavelength, showed percentages relative to glass at different spinning times: 45–70% for 120 min, 35–73% for 30 min, 65–72% for 10 min, 73–92% for 5 min, and 69–88% for 3 min, respectively (Figure 1B). When each gelNF membrane was placed over text, the text was visible through the membrane; however, it was more clearly visible with the 3, 5, and 10 min spun gelNF membranes compared with the 30 and 120 min spun ones (Figure 1A). The thickness, measured with a Vernier caliper, increased proportionally with the spinning time. Specifically, the thickness was approximately 150 μm for 120 min, less than 20 μm for 10 min, and less than 10 μm for 5 min spinning times. In addition to thickness and transparency, the sample subjected to 5 min of spinning and 5 min of crosslinking exhibited better shape maintenance in the liquid in contrast with the sample spun for 3 min, which displayed a contracted shape (Figure 1A).

To confirm the porosity essential for securing the functionality of the corneal endothelium, as well as to elucidate the advantages of using electrospinning, we utilized SEM to examine the morphology of the fibers and membrane. The results revealed a nanofiber morphology with interconnected pores between nanofibers. Additionally, we observed minimal aggregation of the gelatin solution, predominantly resulting in fine nanofibers (Figure 2A). The calculated fiber diameter was 295 ± 82 nm before crosslinking and slightly increased to 304 ± 62 nm after crosslinking, with no significant differences (Figure 2B). To assess whether the produced gelNF membrane possessed a tensile strength similar to DM and suitable mechanical properties for transplantation, we measured the tensile strength and calculated the elastic modulus by converting the data into a stress–strain curve. The fabricated gelNF membranes showed enough mechanical properties to measure the tensile strength. The elastic modulus of the gelNF membrane was 0.4 ± 0.15 MPa, which has no significant differences with human DM (Figure 2C). The permeability of the gelNF membrane, another critical function, was assessed using FITC–dextran. In the gelNF membrane cultured with CECs, FITC–dextran permeated through the membrane at concentrations of 0.286 ± 0.002 μg/mL for 10 kDa and 0.13 ± 0.003 μg/mL for 70 kDa (Figure 3A). Analyzing the degradability of the gelNF membrane through changes in thickness and permeability, when incubated in PBS for 14 days, the initial 6 μm thickness of the gelNF membrane decreased to less than 1 μm, showing an 88.6% thickness reduction after 14 days (Figure 3C). Additionally, beginning at 7 days, there was a significant increase in permeability for the 10 kDa dextran (Figure 3B).

### 3.2. IhCEC Culture on Top of the gelNF Membrane

The cytotoxicity assessment of the gelNF membrane using the Live & Dead assay on IhCECs demonstrated predominant cell attachment in a viable state on both days 3 and 7 (Appendix A). The calculated live cell ratios indicated that 93% and 97% of IhCECs remained viable on days 3 and 7, respectively (Appendix A). Following the verification of the suitable characteristics of the gelNF membrane as a corneal graft material, CECs were cultured to assess cellular compatibility. IhCECs cultured in the gelNF showed comparable proliferation to TCP after 24 h (Figure 4A), and it was confirmed that high-density culture was possible up to day 7 (Figure 4B). Furthermore, the immunofluorescence staining of the similar morphology and expression levels of functional proteins such as ZO-1 and Na^+^/K^+^-ATPase in IhCECs cultured in the gelNF membrane were comparable to those cultured in TCP (Figure 4C).

### 3.3. PrCEC Isolation and Cultivation on Top of the gelNF Membrane

After confirming the appropriate cultivation of immortalized cells in the gelNF membrane, PrCECs were isolated from rabbit eyeballs to confirm the primary cell compatibility. We investigated whether the cells exhibited a hexagonal shape, a characteristic feature of CECs, and expressed junctional proteins. Utilizing phase-contrast imaging on day 4, it was observed that PrCECs formed a monolayer with a high cell density (Appendix A). Additionally, PrCECs were observed to proliferate initially, and from day 3, they became tightly packed with a decreased elongation ratio, forming a normal hexagonal shape and junctional proteins (Appendix A). Furthermore, when PrCECs were cultured in the gelNF membrane for one day, phase-contrast imaging revealed the absence of detached cells, and Live & Dead staining indicated that most cells were viable (Figure 5A). Immunofluorescence staining for functional protein expression showed that comparable to the control FNC-coating-mix-coated coverslips, CECs on the gelNF membrane gradually formed tight junctions from day 3, becoming more pronounced by day 7. Adherent junctional proteins and aquaporin-1 were also expressed in a similar pattern to the coverslip substrate (Figure 5B). The expression of the sodium–potassium ion pump appeared to increase with the progression of the culture (Appendix A).

### 3.4. Cultured CEC Transplantation to the Ex Vivo Rabbit Eyes

The gelNF membrane exhibited mechanical properties comparable to DM and demonstrated a folding shape akin to that observed in the glass Geuder cannula (Figure 6A). In the CEC-cultured gelNF membrane transplantation test, the Geuder cannula was used to insert an 8 mm disc of rolled gelNF membrane with CECs into an artificial anterior chamber. A marking system of peripheral lines was employed to ensure the correct orientation of the cells upon unrolling. It was observed that the membrane unfolded with ease. Air was then injected underneath the membrane to facilitate its adherence to the posterior corneal stroma. Following 5 min of air bubble incubation, the stable attachment of the membrane to the cornea after the addition of the culture medium was confirmed (Figure 6B). Throughout a 3-day culture period and subsequent fixation and washing steps, the membrane remained firmly adhered to the cornea (Figure 6C). Immunofluorescence staining of the phalloidin, ZO-1, and nuclei in the transplanted CEC-cultured gelNF membrane demonstrated a lower density of stromal cells in the cornea with the DM removed. Conversely, in the region where the gelNF membrane was transplanted, a higher density of CECs (nuclei and phalloidin) was confirmed. (Figure 6D,E). The gelNF membrane exhibited autofluorescence, allowing the confirmation of ZO-1 expression (white arrows) via partial z-stacked imaging (Figure 6F).

## 4. Discussion

The primary role of the corneal endothelium is to maintain cornea clarity and thickness by regulating the movement of nutrients and water through the cornea via cell–cell tight junctions and the action of the Na^+^/K^+^-ATPase pump [2,3,4,5]. To preserve this critical function, any material used for CEC transplantation must be permeable. The aim of our research was to design a carrier for cultivated CECs that optimizes transparency and thus visual acuity without reducing efficient permeability, a vital aspect of CEC functionality. Consequently, we analyzed the membrane's thickness for maintaining CEC functionality, permeability, cell compatibility, and transplantability to assess its suitability. First, we compared the thickness and transparency of the gelNF membranes produced at different spinning and crosslinking times to optimize the electrospinning parameters. We found thin membranes undergo rapid and excessive crosslinking, which influences transparency. Membranes that are too thick also exhibit reduced transparency. In our work, the ideal transparency was achieved by utilizing gelNF membranes crafted via a 5 min spinning followed by 5 min crosslinking procedure (Figure 1). The glutaraldehyde (GA) induces crosslink formation between the amine and lysine moieties of gelatin, impacting not only its color but also transparency [14,15]. GA crosslinking is an efficient crosslinking method for electrospun gelNF membranes with a short crosslinking time. However, there is still a risk of toxicity associated with using GA. In studies utilizing the GA vapor crosslinking system, exposure for several hours did not pose toxicity issues [16,17,18]. In the current research, after performing GA crosslinking for only 5 min, a process was carried out to remove residual reagent over a sufficient period, exhibiting good biocompatibility (Figure 4A and Figure 5A, and Appendix A).

Gelatin is a substance derived from collagen and is utilized in various research fields due to its high biocompatibility [18,19]. There have been studies on producing nanofibers using electrospinning, mostly blending them with synthetic polymers to enhance their mechanical properties. However, existing studies using natural polymers have highlighted the potential for residual toxic solvent because of the increased crosslinking level required due to their inadequate biomechanical properties [20,21]. While this approach can achieve high tensile strength, the use of organic solvents may lead to toxicity issues. For the gelNF membrane used in this experiment, vapor crosslinking was carried out using a GA solution, which is the least toxic method for gelatin crosslinking. In the tensile strength test, although there is an approximately eight-fold difference in properties compared with the DM of the human cornea (Figure 2B), when compared with the overall tensile strength of the cornea (15.8 MPa) [22,23], it can be considered similar to DM and deemed suitable as a gelNF membrane to fulfill the role of Descemet’s membrane. The mechanical properties of the endothelium graft are required to be appropriate to endure the stretching forces and maintain integrity during surgery and post-transplantation as the cornea goes through various movements and stresses. Moreover, an optimal balance of tensile strength is necessary to ensure that the material is strong enough to maintain its structural integrity and support the endothelial cells while also allowing for necessary flexibility and compliance within the corneal environment. In this study, the developed gelNF membrane demonstrated suitable properties for transplantation (Figure 2B and Figure 6A) and appropriate degradation rates (Figure 3C), addressing the limitations of materials that degrade too rapidly during cell cultivation while maintaining the desired properties for CEC delivery.

Thickness is one of the crucial considerations for a material intended for transplantation into the corneal endothelium. It is important to have a thickness comparable to native DM. The gelNF membrane used in this study was optimized at less than 20 μm after 5 min of spinning. Considering that carriers utilized for DMEK should be less than 50 μm [24], the gelNF membrane fabricated in this study is adequate for use as a CEC transplantation material. The other advantage of an electrospun nanofiber membrane is its ability to ensure permeability via pore structures, as reported in various studies [7,25,26,27]. In this study, the porosity of the gelNF membrane produced can be observed in the pore structures between the nanofibers in the top view image obtained via SEM (Figure 2A). Additionally, a porous structure can be confirmed in the cross-sectioned view image (Figure 3C, day 0). The permeability test using dextran is an analytical method primarily utilized in membranes that function as barriers [28]. In conjunction with SEM analysis, permeability experiments were conducted using FITC-labeled dextran to ascertain the extent to which actual molecules penetrated the gelNF membrane. In this study, both 10 kDa and 70 kDa dextran was observed to permeate the gelNF membrane, indicating its permeability property (Figure 3A). Permeability was also associated with degradation. A significant difference in permeability was noted seven days after incubating the membrane at body temperature (Figure 3B). Analysis of the thickness changes in the gelNF membrane revealed that over 88% of the membrane degraded after incubation at body temperature for 14 days (Figure 3C). The rapid degradation of the gelNF membrane may be advantageous in terms of eliminating potential factors that can induce an inflammatory response within the eye. However, further validation is needed to determine if this timeframe is sufficient to achieve the primary objective of CEC delivery. Indirectly interpreting the ex vivo tests conducted in this study, it is presumed that CECs would have begun migrating from the gelNF membrane to the cornea, considering that the gelNF membrane had already attached to the posterior cornea after 5 min of incubation following graft transplantation (Figure 6B).

To maintain the thickness and transparency of the cornea, preserving the pump function of CECs is crucial. To achieve this, it is important to maintain a high cell density of CECs while forming tight junctions. In this study, two types of CECs were tested. Firstly, it was confirmed that the gelNF membrane allows for the high-cell-density cultivation of IhCECs without cytotoxicity (Figure 4A,B). Functional protein expression confirmed using immunofluorescence staining revealed patterns comparable to the control group in this study, represented in the coverslip, showing the expressions of ZO-1 and sodium–potassium pump ions. Likewise, PrCECs cultured on the gelNF membrane showed high cell attachment without cytotoxicity from day 1, as well as a clearer hexagonal cell shape (Figure 5A). Furthermore, the functional protein expression was comparable to that of CECs cultured on coverslips (Figure 5B). These results of a high CEC density and functional protein expression demonstrate the gelNF membrane is suitable for CEC cultivation.

Lastly, an ex vivo study was conducted to evaluate the practical application of the CEC-cultured gelNF membrane. The artificial anterior chamber, considered an excellent tool for DMEK surgery wet labs [29,30], was utilized in this experiment to assess the material prior to transplantation in an animal model. Grafts inserted into the anterior chamber through a glass Geuder cannula were affixed to the posterior of the cornea using an air bubble for only 5 min (Figure 6B). This attachment occurred much faster than the several hours or days of gas bubble use with supine positioning in current DMEK procedures [31]. While the reasons for this relatively rapid adhesion are not fully known, it is likely a result of the high surface contact area as a result of the mesh-like surface topology of the gelatin nanofibers. While further work is merited to investigate this phenomenon in live animals (and patients), this potentially reduced adhesion time presents the advantages of reduced patient discomfort with strict supine positioning in the post-operative period and eliminating the need for re-bubbling procedures. Subsequent examination of the cytoskeleton (phalloidin) and tight junction protein expression in CECs post-transplantation revealed sustained expression for three days when in contact with both the gelNF membrane and the cornea (Figure 6E,F). Despite the autofluorescence of the gelNF membrane, the clear visualization of ZO-1 expression was aided by employing partial z-stack imaging to confirm tight junction protein formation (Figure 6F, white arrows). In this study, an engraftment test was conducted using rabbit corneas over a period of three days. As validated in the author’s prior research [32], it was observed that corneal endothelial cells migrated to decellularized cornea within three days of cultivation. Consequently, it is anticipated that during the degradation period of the membrane, there is ample time for corneal endothelial cells to migrate to the cornea. To validate this accurately, the transplantation results need confirmation via subsequent studies. However, considering that the membrane adhered to the cornea during the washing and fixation processes after three days of cultivation, it is believed that interactions may have occurred between cells and corneal tissue or between the gelNF membrane and corneal tissue.

There are aspects that need to be validated via future research. While qualitative validation of functionality was performed by culturing CECs on the gelNF membrane and then transplanting them after ex vivo study, there is a need for quantitative analysis as well. Additionally, it is imperative to verify the inflammatory response in the body via in vivo testing. Furthermore, using CEC-cultured gelNF membrane transplantation in disease models, such as treating edematous cloudy corneas due to corneal endothelium dysfunction, will further demonstrate its viability as a substitute for donor transplant surgery.

## 5. Conclusions

The objective of this study was to develop a material for transplanting cultured CECs. A gelNF membrane was developed using electrospinning that maintained sufficient thickness, transparency, and permeability for proper corneal functionality. We also confirmed that CECs growth on the gelNF membrane expressed functional biomarkers such as ZO-1. Additionally, via ex vivo studies, it was verified that the CECs cultured on the gelNF membrane exhibited properties suitable for transplantation and successfully adhered to the cornea. Therefore, the gelNF membrane demonstrates potential as a material for cultured CEC transplantation.

## Figures and Tables

**Figure 1 bioengineering-11-00054-f001:**
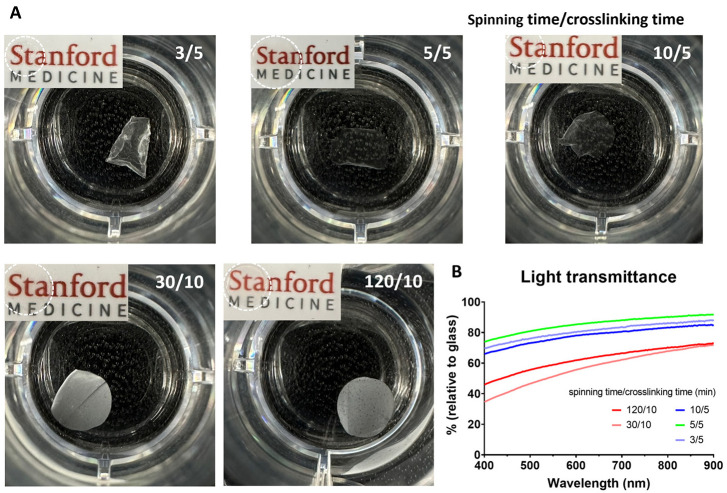
Optimization of spinning time and crosslinking time based on thickness and transparency of the gelNF membrane. (**A**) Macroscopic image of gelNF membrane under varied spinning and crosslinking times. The numbers on the top right of the image represent spinning time on the left of the slash and crosslinking time on the right of the slash. The number in the top right corner indicates spinning time/GA crosslinking time. (**B**) Transparency measurements of gelNF membranes made with different spinning and crosslinking times as a function of wavelength.

**Figure 2 bioengineering-11-00054-f002:**
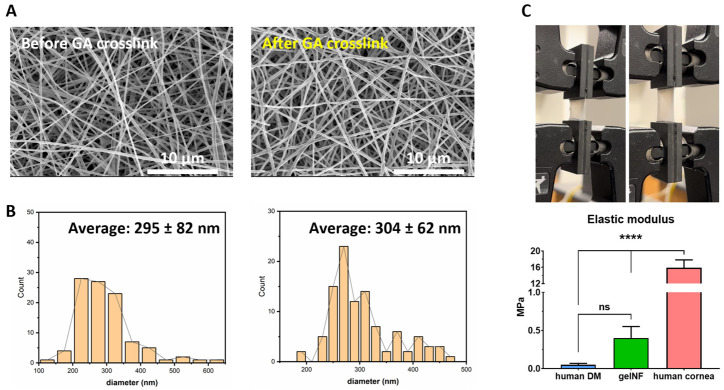
Analysis of nanofiber diameter and mechanical properties of gelNF membrane. (**A**) SEM images of the gelNF membrane before/after GA crosslinking and (**B**) calculation of the nanofiber diameter. (**C**) Tensile stress test of gelNF membrane and calculated elastic modulus, elongation at break, and ultimate tensile strength. ns = no significant differences and **** = significant differences.

**Figure 3 bioengineering-11-00054-f003:**
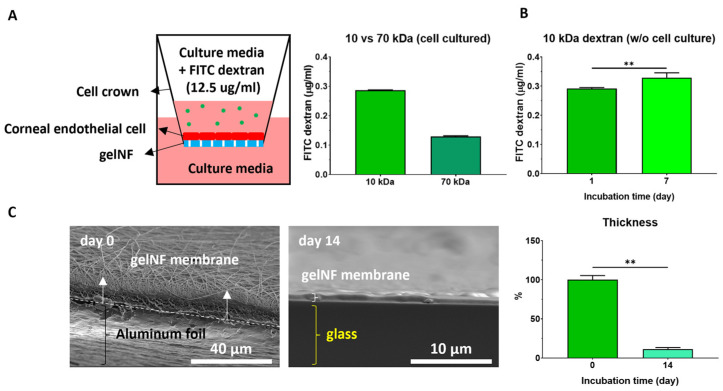
Permeability and degradability of the gelNF membrane. (**A**) Permeability test of the gelNF membrane using FITC–dextran and cell insert. (**B**) Measurement of permeability changes in the gelNF membrane over incubation time. Day 0 scale bar: 40 µm, and day 14 scale bar: 10 µm. (**C**) Measurement of the gelNF membrane degradation using changes in thickness. ** = significant differences.

**Figure 4 bioengineering-11-00054-f004:**
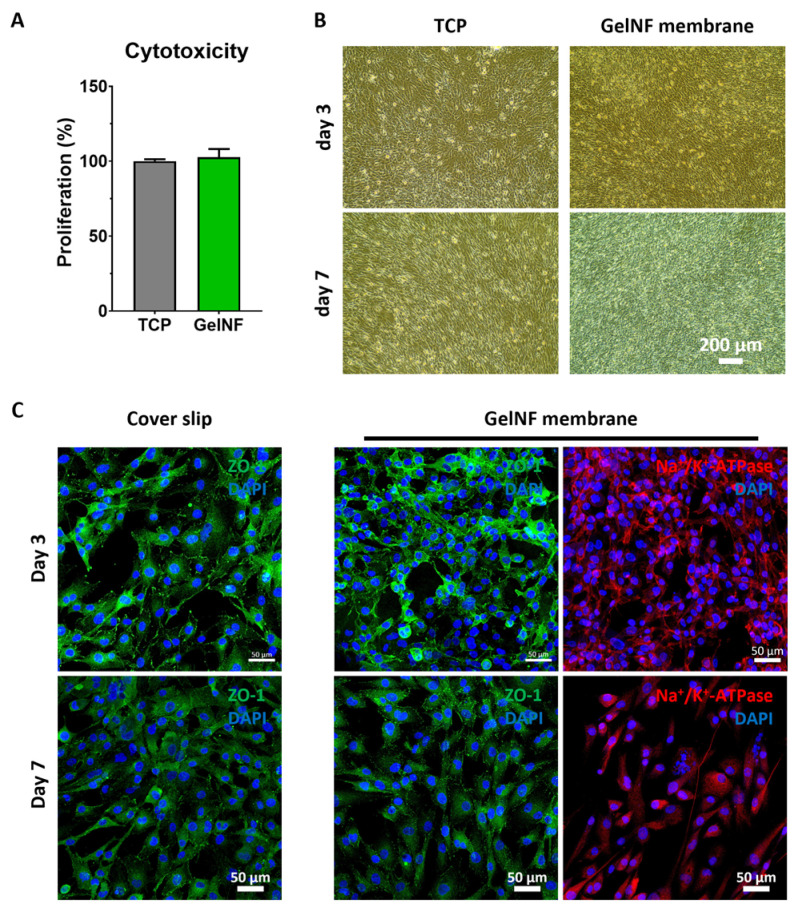
IhCEC culture on top of the gelNF membrane. (**A**) Cytotoxicity of the gelNF membrane relative to the TCP and (**B**) comparison of the morphology and cell density of the IhCECs cultured on TCP and the gelNF membrane on days 3 and 7. Scale bar: 200 µm. (**C**) Immunofluorescence staining of tight junction proteins (ZO-1), pump protein (Na^+^/K^+^-ATPase), and nuclei (DAPI) of the IhCECs on days 3 and 7. Scale bars: 50 µm.

**Figure 5 bioengineering-11-00054-f005:**
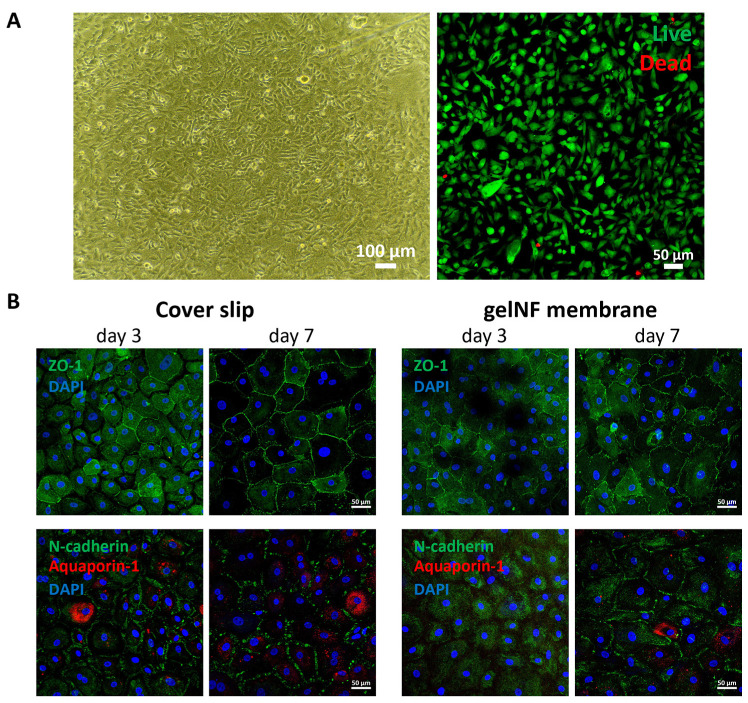
PrCEC culture on top of the gelNF membrane. (**A**) Morphology and Live & Dead staining of the PrCECs cultured on top of the gelNF membrane on day 1. Scale bar in phase-contrast image: 100 µm, and for the Live & Dead staining: 50 µm. (**B**) Immunofluorescence staining of the junctional proteins (ZO-1 and N-cadherin), water pump ion (aquaporin-1), and nuclei (DAPI) on days 3 and 7. Scale bars: 50 µm.

**Figure 6 bioengineering-11-00054-f006:**
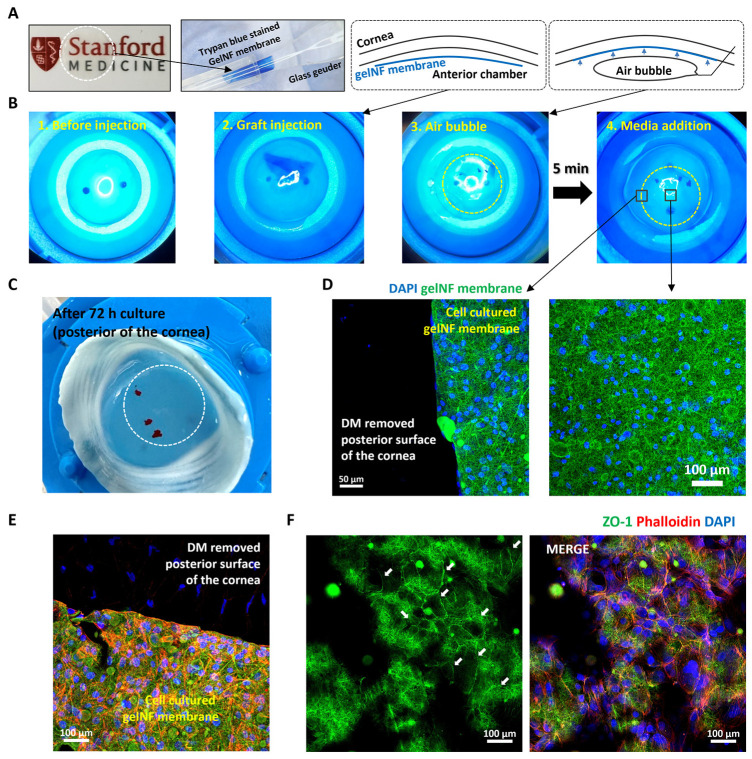
Ex vivo study for transplantation of PrCECs cultured on top of the gelNF membrane using Geuder and artificial anterior chamber. (**A**) Macro-images of gelNF membrane placed inside of the Geuder and schematic images of PrCECs cultured on top of the transplanted gelNF membrane. (**B**) Macro-images of (1) preparation of DM and endothelium removed from artificial anterior chamber, (2) after the graft injection, (3) air bubble applied to unfold the membrane with 5 min incubation, and (4) addition of culture medium. (**C**) Macro-image of the posterior of the cornea indicates that gelNF membrane remained on the cornea (after 72 h of culture). (**D**) Immunofluorescence staining of transplanted PrCECs and gelNF membrane after 72 h. GelNF membrane showed autofluorescence to green, and the nuclei of the PrCECs were labeled with DAPI. Scale bars: 50 µm (**left**) and 100 µm (**right**). (**E**) Immunofluorescence staining of ZO-1 (green), phalloidin (red), and nuclei (blue) transplanted PrCEC-cultured gelNF membrane. The image was obtained from the edge of the gelNF membrane. (**F**) Partially Z-stacked images to visualize the ZO-1 expression (white arrows) that was obscured by autofluorescence of the gelNF membrane. Scale bars: 50 µm.

## Data Availability

The datasets created and/or analyzed during the current investigation are available upon reasonable request from the corresponding author.

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
