# Peer review of "Electrospun Nanofiber Membrane for Cultured Corneal Endothelial Cell Transplantation"

_bioengineering, 2024, doi:10.3390/bioengineering11010054_

Round 1
Reviewer 1 Report (Previous Reviewer 1)
Comments and Suggestions for Authors
The authors answered to all my questions and comments and I recommend this manuscript for acceptance.
I suggest to add the information and explanation about migration of endothelial cells from biomaterial to the acellular cornea after 3 days after transplantation from previous research to the manuscript. It will help to more understand the experiments and manuscript. I also please authors to add the information about the length of the experiments into the figure legends. Sometimes it is there, sometimes not. Since it differs, also for the experiments in one figure, it will be easier for readers to understand without searching through the text.
Author Response
Please see the attachment

Reviewer 2 Report (Previous Reviewer 2)
Comments and Suggestions for Authors
The transparency of gelNF membrane is suggested to be improved in the future.
Author Response
Please see the attachment

Reviewer 3 Report (Previous Reviewer 3)
Comments and Suggestions for Authors
Accepted after calrification for all comments
Author Response
Thank you very much
Reviewer 4 Report (Previous Reviewer 4)
Comments and Suggestions for Authors
I recommend to accept.
Author Response
Thank you very much
This manuscript is a resubmission of an earlier submission. The following is a list of the peer review reports and author responses from that submission.
Round 1
Reviewer 1 Report
Comments and Suggestions for Authors
The authors are working on highly actual topic, which is nowadays very important and will be even more important in next years.
My main questions and suggestions to the presented manuscript:
1. In manuscript is used for cross-linking glutaraldehyde. It was published, that glutaraldehyde cross-linked collagen films increased apoptosis of cells (Gough et al., 2002). The use of cytotoxic glutaraldehyde in the product suitable for transplantation is questionable. Can authors hypothesise about it and compare their results with previous?
2. Line 144, the methodology of 2.6 part should be written more precisely. It is not clear which type of cells is used (human/ rabbit) and which cornea (human/rabbit). It is confusing, as authors were using various types of cells during the project. Same unspecified description is in results part. Only in Figure legend is description correct. More specific explanation will help readers with understanding the experiments.
3. How many repeats did the experiment 2.6 have? I am missing this detail in the methodology. Have the repeated experiments had the same results as it is shown on Figure 6? IF yes, please mention it in results and discussion part.
4. In 2.6 part is missing information, how long were cells cultured on gelNF before implantation.
5. The degradation of the biomaterial is very quick (88% in 14 days). I have doubts about using this material as DMEK replacement with such a quick degradation. Could authors hypothesise about it?
6. What are the effects of the degraded biomaterial in the cornea/eye? How big molecules are released into the anterior chamber or other parts of the eye?
7. According previous studies of the cytotoxicity of the glutaraldehyde, I consider the measurement of the cytotoxicity just after 24 hours as little evidence. As this question is very important, I suggest to prolong the experiment at least on 7 days, same as other experiments in Figure 4. Authors showed IHF staining after 7 days, but not the cytotoxicity results. The test of apoptosis would be very helpful.
8. The quality of the picture from immunofluorescence staining in Fig 4 after 7 days, Na+/K+ ATPase, is poor. Could you please use better image?
9. How you can explain the difference between immunofluorescence staining of ZO-1 protein in Figure 4 and Figure 5, 6? In Figure 4 is not stained just the cell membrane, but signal is also in cytoplasm.
10. In Figure 5 is shown the Live/Dead staining after 24 hours. According the risk of glutaraldehyde cytotoxicity, the longer time should be use (7 days) for proving the biocompatibility of the gelNF.
11. Use gelNF as a DMEK replacement means longer treatment in the patient eye. Authors showed cells with endothelial morphology after 7 days of culture. In the implantation experiment (Figure 6), authors showed results only after 3 days. Why was not used longer, the 7-day interval?
12. The Data availability statement is completely missing!
Minor questions and suggestions to the presented manuscript:
1. In the introduction, line 35, the shortcut CECs is not explained.
2. Line 124, centrifuged is wrongly spelled.
3. Line 125, the sentence doesn´t make sense to me. Probably should be connected to previous one.
4. Line 157, the expression ”function protein expression” is not understandable.
5. In Figure 3, the scale bar is not clearly visible.
6. Line 284, “the cornea with D removed.” Maybe DM should be there.
7. In discussion is used GA shortcut which was not explained before.
Reviewer 2 Report
Comments and Suggestions for Authors
The present manuscript titled “Electrospun nanofiber membrane for cultured corneal endothelial cell transplantation”, introduces the research of electrospinning nanofiber in corneal endothelial transplantation. Through improved process parameters and comparative analysis of experiments, the advantages and potential of gelNF membrane developed by electrospinning as a CECs transplantation material are demonstrated. It provides inspiration for the design and preparation of corneal transplantation materials with clinical application. Finally, it points out the shortcomings of the current research and gives a prospect. The content of the article is detailed and full of data, but it still needs further improvement carefully. Here are some specific suggestions:
1) Article 3.1 describes that the transparency display gradients of gelNF films measured at visible wavelengths are 45-70% for 120 min, 35-73% for 30 min, 65-72% for 10 min, and 73-92% for 5 min, respectively. The transparency description at 3 minutes is missing, please add it.
2) The phenomenon that the gelNF film rotating for 3 minutes does not unfold correctly as stated at the end of the first paragraph of article 3.1 is inconsistent with the conclusion that the thinner film has a shorter spinning time and is more easily formed in liquid. How to make the expression clear before and after is subject to the author's consideration?
3) The vertical axis diagram of Figure 1B in this article shows the incorrect representation of the transparency of gelNF film at visible wavelengths. The content in parentheses should be %.
4) The explanation of Figure 2 in the article lacks the explanation of Figure 2C, which needs to be supplemented by the author.
5) According to the second paragraph of article 3.1, it can be seen from Figure 2C that the elastic modulus of gelNF film is 0.4± 0.15MPa, which is not significantly different from that of human DM. The comparison between human cornea and geINF in Figure 2C is missing, please consider.
6) The contents of two paragraphs in article 3.2 are similar, and there are mistakes in the expression of repetition. Please correct them.
7) The role of Figure 6E is not explained in article 3.4, please consider it.
8) The references on electrospinning may be considered: Macromolecular Bioscience, 2023, 23, 2300105; Journal of Polymer Research, 2021, 28: 232. Adsorption Science & Technology 2019, 37(5-6): 412–424.
Comments on the Quality of English LanguagePlease pay attention to the consistency of grammar, tense, etc.
Reviewer 3 Report
Comments and Suggestions for Authors
The manuscript is suitable for publication after addressing the following comments.
1. The introduction part should be written to include the selection for gelataion as a source for nanofiber, advantage and disadvantage.
Mechanical properties for the formed nanofiber should be performed.
In addition, for good application, contact angle should be assessed.
What is the physical difference between the two nanofibers (Before GA crosslink and after GA crosslink)?
Reviewer 4 Report
Comments and Suggestions for Authors
This manuscript manufactured a gelatin nanofiber membrane using electrospinning, and its structure, properties and cell response were investigated in detail, providing a potential substitute for corneal endothelium. There are a few comments that need to be considered:
1. In figure 1, the labels of each group (‘3/5’, ‘5/5’, etc.) need to be explained in the manuscript or figure captions.
2. Which fabrication parameters did the authors use for the mechanical and permeability tests?
3. Please add the caption of figure 2C.
